# Curvature and Optimal Algorithms for Learning and Minimizing Submodular Functions

**Rishabh Iyer**[†]**, Stefanie Jegelka**[∗]**, Jeff Bilmes**[†]
[†] University of Washington, Dept. of EE, Seattle, U.S.A.
[∗] University of California, Dept. of EECS, Berkeley, U.S.A.
rkiyer@uw.edu, stefje@eecs.berkeley.edu, bilmes@uw.edu

## Abstract

We investigate three related and important problems connected to machine learning: approximating a submodular function everywhere, learning a submodular function (in a PAC-like setting [28]), and constrained minimization of submodular functions. We show that the complexity of all three problems depends on the "curvature" of the submodular function, and provide lower and upper bounds that refine and improve previous results [2, 6, 8, 27]. Our proof techniques are fairly generic. We either use a black-box transformation of the function (for approximation and learning), or a transformation of algorithms to use an appropriate surrogate function (for minimization). Curiously, curvature has been known to influence approximations for submodular maximization [3, 29], but its effect on minimization, approximation and learning has hitherto been open. We complete this picture, and also support our theoretical claims by empirical results.

## 1  Introduction

Submodularity is a pervasive and important property in the areas of combinatorial optimization, economics, operations research, and game theory. In recent years, submodularity's use in machine learning has begun to proliferate as well. A set function $f : 2^V \rightarrow \mathbb{R}$ over a finite set $V = \{1, 2, \dots, n\}$ is *submodular* if for all subsets $S, T \subseteq V$, it holds that $f(S) + f(T) \geq f(S \cup T) + f(S \cap T)$. Given a set $S \subseteq V$, we define the *gain* of an element $j \notin S$ in the context $S$ as $f(j|S) \triangleq f(S \cup j) - f(S)$. A function $f$ is submodular if it satisfies *diminishing marginal returns*, namely $f(j|S) \geq f(j|T)$ for all $S \subseteq T, j \notin T$, and is *monotone* if $f(j|S) \geq 0$ for all $j \notin S, S \subseteq V$.

While submodularity, like convexity, occurs naturally in a wide variety of problems, recent studies have shown that in the general case, many submodular problems of interest are very hard: the problems of learning a submodular function or of submodular minimization under constraints do not even admit constant or logarithmic approximation factors in polynomial time [2, 7, 8, 10, 27]. These rather pessimistic results however stand in sharp contrast to empirical observations, which suggest that these lower bounds are specific to rather contrived classes of functions, whereas much better results can be achieved in many practically relevant cases. Given the increasing importance of submodular functions in machine learning, these observations beg the question of qualifying and quantifying properties that make sub-classes of submodular functions more amenable to learning and optimization. Indeed, limited prior work has shown improved results for constrained minimization and learning of sub-classes of submodular functions, including symmetric functions [2, 25], concave functions [7, 18, 24], *label cost* or covering functions [9, 31].

In this paper, we take additional steps towards addressing the above problems and show how the generic notion of the *curvature* – the deviation from modularity– of a submodular function determines both upper and lower bounds on approximation factors for many learning and constrained optimization problems. In particular, our quantification tightens the generic, function-independent bounds in [8, 2, 27, 7, 10] for many practically relevant functions. Previously, the concept of curvature has been used to

tighten bounds for submodular maximization problems [3, 29]. Hence, our results complete a unifying picture of the effect of curvature on submodular problems. Moreover, curvature is still a fairly generic concept, as it only depends on the marginal gains of the submodular function. It allows a smooth transition between the 'easy' functions and the 'really hard' subclasses of submodular functions.

## 2 Problem statements, definitions and background

Before stating our main results, we provide some necessary definitions and introduce a new concept, the *curve normalized* version of a submodular function. Throughout this paper, we assume that the submodular function $f$ is defined on a ground set $V$ of $n$ elements, that it is nonnegative and $f(\emptyset) = 0$. We also use normalized *modular* (or additive) functions $w : 2^V \to \mathbb{R}$ which are those that can be written as a sum of weights, $w(S) = \sum_{i \in S} w(i)$. We are concerned with the following three problems:

**Problem 1.** *(Approximation [8]) Given a submodular function $f$ in form of a value oracle, find an approximation $\hat{f}$ (within polynomial time and representable within polynomial space), such that for all $X \subseteq V$, it holds that $\hat{f}(X) \leq f(X) \leq \alpha_1(n)\hat{f}(X)$ for a polynomial $\alpha_1(n)$.*

**Problem 2.** *(PMAC-Learning [2]) Given i.i.d training samples $\{(X_i, f(X_i)\}_{i=1}^m$ from a distribution $\mathcal{D}$, learn an approximation $\hat{f}(X)$ that is, with probability $1 - \delta$, within a multiplicative factor of $\alpha_2(n)$ from $f$.*

**Problem 3.** *(Constrained optimization [27, 7, 10, 16]) Minimize a submodular function $f$ over a family $\mathcal{C}$ of feasible sets, i.e., $\min_{X \in \mathcal{C}} f(X)$.*

In its general form, the approximation problem was first studied by Goemans et al. [8], who approximate any monotone submodular function to within a factor of $O(\sqrt{n} \log n)$, with a lower bound of $\alpha_1(n) = \Omega(\sqrt{n}/\log n)$. Building on this result, Balcan and Harvey [2] show how to PMAC-learn a monotone submodular function within a factor of $\alpha_2(n) = O(\sqrt{n})$, and prove a lower bound of $\Omega(n^{1/3})$ for the learning problem. Subsequent work extends these results to sub-additive and fractionally sub-additive functions [1]. Better learning results are possible for the subclass of *submodular shells* [23] and Fourier sparse set functions [26]. Both Problems 1 and 2 have numerous applications in algorithmic game theory and economics [2, 8] as well as machine learning [2, 22, 23, 26, 15].

Constrained submodular minimization arises in applications such as power assignment or transportation problems [19, 30, 13]. In machine learning, it occurs, for instance, in the form of MAP inference in high-order graphical models [17] or in size-constrained corpus extraction [21]. Recent results show that almost all constraints make it hard to solve the minimization even within a constant factor [27, 6, 16]. Here, we will focus on the constraint of imposing a lower bound on the cardinality, and on combinatorial constraints where $\mathcal{C}$ is the set of all $s$-$t$ paths, $s$-$t$ cuts, spanning trees, or perfect matchings in a graph.

A central concept in this work is the total *curvature* $\kappa_f$ of a submodular function $f$ and the curvature $\kappa_f(S)$ with respect to a set $S \subseteq V$, defined as [3, 29]

$$\kappa_f = 1 - \min_{j \in V} \frac{f(j \mid V \setminus j)}{f(j)}, \qquad \kappa_f(S) = 1 - \min_{j \in S} \frac{f(j|S\setminus j)}{f(j)}. \tag{1}$$

Without loss of generality, assume that $f(j) > 0$ for all $j \in V$. It is easy to see that $\kappa_f(S) \leq \kappa_f(V) = \kappa_f$, and hence $\kappa_f(S)$ is a tighter notion of curvature. A modular function has curvature $\kappa_f = 0$, and a matroid rank function has maximal curvature $\kappa_f = 1$. Intuitively, $\kappa_f$ measures how far away $f$ is from being *modular*. Conceptually, curvature is distinct from the recently proposed *submodularity ratio* [5] that measures how far a function is from being *submodular*. Curvature has served to tighten bounds for submodular maximization problems, e.g., from $(1-1/e)$ to $\frac{1}{\kappa_f}(1-e^{-\kappa_f})$ for monotone submodular maximization subject to a cardinality constraint [3] or matroid constraints [29], and these results are tight. For submodular minimization, learning, and approximation, however, the role of curvature has not yet been addressed (an exception are the upper bounds in [13] for minimization). In the following sections, we complete the picture of how curvature affects the complexity of submodular maximization and minimization, approximation, and learning.

The above-cited lower bounds for Problems 1–3 were established with functions of maximal curvature ($\kappa_f = 1$) which, as we will see, is the worst case. By contrast, many practically interesting functions have smaller curvature, and our analysis will provide an explanation for the good empirical results

observed with such functions [13, 22, 14]. An example for functions with $\kappa_f < 1$ is the class of concave over modular functions that have been used in speech processing [22] and computer vision [17]. This class comprises, for instance, functions of the form $f(X) = \sum_{i=1}^{k}(w_i(X))^a$, for some $a \in [0, 1]$ and a nonnegative weight vectors $w_i$. Such functions may be defined over clusters $C_i \subseteq V$, in which case the weights $w_i(j)$ are nonzero only if $j \in C_i$ [22, 17, 11].

**Curvature-dependent analysis.** To analyze Problems $1-3$, we introduce the concept of a *curve-normalized* polymatroid[1]. Specifically, we define the $\kappa_f$-*curve-normalized* version of $f$ as

$$f^{\kappa}(X) = \frac{f(X) - (1 - \kappa_f)\sum_{j \in X} f(j)}{\kappa_f} \tag{2}$$

If $\kappa_f = 0$, then we set $f^{\kappa} \equiv 0$. We call $f^{\kappa}$ the curve-normalized version of $f$ because its curvature is $\kappa_{f^{\kappa}} = 1$. The function $f^{\kappa}$ allows us to decompose a submodular function $f$ into a "difficult" polymatroid function and an "easy" modular part as $f(X) = f_{\text{difficult}}(X) + m_{\text{easy}}(X)$ where $f_{\text{difficult}}(X) = \kappa_f f^{\kappa}(X)$ and $m_{\text{easy}}(X) = (1 - \kappa_f)\sum_{j \in X} f(j)$. Moreover, we may modulate the curvature of given any function $g$ with $\kappa_g = 1$, by constructing a function $f(X) \triangleq cg(X) + (1 - c)|X|$ with curvature $\kappa_f = c$ but otherwise the same polymatroidal structure as $g$.

Our curvature-based decomposition is different from decompositions such as that into a *totally normalized* function and a modular function [4]. Indeed, the curve-normalized function has some specific properties that will be useful later on (proved in [12]):

**Lemma 2.1.** *If $f$ is monotone submodular with $\kappa_f > 0$, then $f(X) \leq \sum_{j \in X} f(j)$ and $f(X) \geq (1 - \kappa_f)\sum_{j \in X} f(j)$.*

**Lemma 2.2.** *If $f$ is monotone submodular, then $f^{\kappa}(X)$ in Eqn. (2) is a monotone non-negative submodular function. Furthermore, $f^{\kappa}(X) \leq \sum_{j \in X} f(j)$.*

The function $f^{\kappa}$ will be our tool for analyzing the hardness of submodular problems. Previous information-theoretic lower bounds for Problems 1–3 [6, 8, 10, 27] are *independent* of curvature and use functions with $\kappa_f = 1$. These curvature-independent bounds are proven by constructing two essentially indistinguishable matroid rank functions $h$ and $f^R$, one of which depends on a random set $R \subseteq V$. One then argues that any algorithm would need to make a super-polynomial number of queries to the functions for being able to distinguish $h$ and $f^R$ with high enough probability. The lower bound will be the ratio $\max_{X \in \mathcal{C}} h(X)/f^R(X)$. We extend this proof technique to functions with a fixed given curvature. To this end, we define the functions

$$f^R_{\kappa}(X) = \kappa_f f^R(X) + (1 - \kappa_f)|X| \quad \text{and} \quad h_{\kappa}(X) = \kappa_f h(X) + (1 - \kappa_f)|X|. \tag{3}$$

Both of these functions have curvature $\kappa_f$. This construction enables us to explicitly introduce the effect of curvature into information-theoretic bounds for all monotone submodular functions.

**Main results.** The curve normalization (2) leads to refined upper bounds for Problems 1–3, while the curvature modulation (3) provides matching lower bounds. The following are some of our main results: for approximating submodular functions (Problem 1), we replace the known bound of $\alpha_1(n) = O(\sqrt{n}\log n)$ [8] by an improved curvature-dependent $O(\frac{\sqrt{n}\log n}{1+(\sqrt{n}\log n-1)(1-\kappa_f)})$. We complement this with a lower bound of $\tilde{\Omega}(\frac{\sqrt{n}}{1+(\sqrt{n}-1)(1-\kappa_f)})$. For learning submodular functions (Problem 2), we refine the known bound of $\alpha_2(n) = O(\sqrt{n})$ [2] in the PMAC setting to a curvature dependent bound of $O(\frac{\sqrt{n}}{1+(\sqrt{n}-1)(1-\kappa_f)})$, with a lower bound of $\tilde{\Omega}(\frac{n^{1/3}}{1+(n^{1/3}-1)(1-\kappa_f)})$. Finally, Table 1 summarizes our curvature-dependent approximation bounds for constrained minimization (Problem 3). These bounds refine many of the results in [6, 27, 10, 16]. In general, our new curvature-dependent upper and lower bounds refine known theoretical results whenever $\kappa_f < 1$, in many cases replacing known polynomial bounds by a curvature-dependent constant factor $1/(1 - \kappa_f)$. Besides making these bounds precise, the decomposition and the curve-normalized version (2) are the basis for constructing tight algorithms that (up to logarithmic factors) achieve the lower bounds.

| Constraint | Modular approx. (MUB) | Ellipsoid approx. (EA) | Lower bound |
|---|---|---|---|
| Card. LB | $\frac{k}{1+(k-1)(1-\kappa_f)}$ | $O\left(\frac{\sqrt{n}\log n}{1+(\sqrt{n}\log n-1)(1-\kappa_f)}\right)$ | $\tilde{\Omega}\left(\frac{n^{1/2}}{1+(n^{1/2}-1)(1-\kappa_f)}\right)$ |
| Spanning Tree | $\frac{n}{1+(n-1)(1-\kappa_f)}$ | $O\left(\frac{\sqrt{m}\log m}{1+(\sqrt{m}\log m-1)(1-\kappa_f)}\right)$ | $\tilde{\Omega}\left(\frac{n}{1+(n-1)(1-\kappa_f)}\right)$ |
| Matchings | $\frac{n}{2+(n-2)(1-\kappa_f)}$ | $O\left(\frac{\sqrt{m}\log m}{1+(\sqrt{m}\log m-1)(1-\kappa_f)}\right)$ | $\tilde{\Omega}\left(\frac{n}{1+(n-1)(1-\kappa_f)}\right)$ |
| s-t path | $\frac{n}{1+(n-1)(1-\kappa_f)}$ | $O\left(\frac{\sqrt{m}\log m}{1+(\sqrt{m}\log m-1)(1-\kappa_f)}\right)$ | $\tilde{\Omega}\left(\frac{n^{2/3}}{1+(n^{2/3}-1)(1-\kappa_f)}\right)$ |
| s-t cut | $\frac{m}{1+(m-1)(1-\kappa_f)}$ | $O\left(\frac{\sqrt{m}\log m}{1+(\log m\sqrt{m}-1)(1-\kappa_f)}\right)$ | $\tilde{\Omega}\left(\frac{\sqrt{n}}{1+(\sqrt{n}-1)(1-\kappa_f)}\right)$ |

Table 1: Summary of our results for constrained minimization (Problem 3).

# 3 Approximating submodular functions everywhere

We first address improved bounds for the problem of approximating a monotone submodular function everywhere. Previous work established $\alpha$-approximations $g$ to a submodular function $f$ satisfying $g(S) \leq f(S) \leq \alpha g(S)$ for all $S \subseteq V$ [8]. We begin with a theorem showing how any algorithm computing such an approximation may be used to obtain a curvature-specific, improved approximation. Note that the curvature of a monotone submodular function can be obtained within $2n + 1$ queries to $f$. The key idea of Theorem 3.1 is to only approximate the curved part of $f$, and to retain the modular part exactly. The full proof is in [12].

**Theorem 3.1.** *Given a polymatroid function $f$ with $\kappa_f < 1$, let $f^\kappa$ be its curve-normalized version defined in Equation (2), and let $\hat{f}^\kappa$ be a submodular function satisfying $\hat{f}^\kappa(X) \leq f^\kappa(X) \leq \alpha(n)\hat{f}^\kappa(X)$, for some $X \subseteq V$. Then the function $\hat{f}(X) \triangleq \kappa_f \hat{f}^\kappa(X) + (1-\kappa_f)\sum_{j\in X} f(j)$ satisfies*

$$\hat{f}(X) \leq f(X) \leq \frac{\alpha(n)}{1+(\alpha(n)-1)(1-\kappa_f)}\hat{f}(X) \leq \frac{\hat{f}(X)}{1-\kappa_f}. \tag{4}$$

Theorem 3.1 may be directly applied to tighten recent results on approximating submodular functions everywhere. An algorithm by Goemans et al. [8] computes an approximation to a polymatroid function $f$ in polynomial time by approximating the submodular polyhedron via an ellipsoid. This approximation (which we call the ellipsoidal approximation) satisfies $\alpha(n) = O(\sqrt{n}\log n)$, and has the form $\sqrt{w^f(X)}$ for a certain weight vector $w^f$. Corollary 3.2 states that a tighter approximation is possible for functions with $\kappa_f < 1$.

**Corollary 3.2.** *Let $f$ be a polymatroid function with $\kappa_f < 1$, and let $\sqrt{w^{f^\kappa}(X)}$ be the ellipsoidal approximation to the $\kappa$-curve-normalized version $\hat{f}^\kappa(X)$ of $f$. Then the function $f^{ea}(X) = \kappa_f\sqrt{w^{f^\kappa}(X)} + (1-\kappa_f)\sum_{j\in X} f(j)$ satisfies*

$$f^{ea}(X) \leq f(X) \leq O\left(\frac{\sqrt{n}\log n}{1+(\sqrt{n}\log n-1)(1-\kappa_f)}\right) f^{ea}(X). \tag{5}$$

If $\kappa_f = 0$, then the approximation is exact. This is not surprising since a modular function can be inferred exactly within $O(n)$ oracle calls. The following lower bound (proved in [12]) shows that Corollary 3.2 is tight up to logarithmic factors. It refines the lower bound in [8] to include $\kappa_f$.

**Theorem 3.3.** *Given a submodular function $f$ with curvature $\kappa_f$, there does not exist a (possibly randomized) polynomial-time algorithm that computes an approximation to $f$ within a factor of $\frac{n^{1/2-\epsilon}}{1+(n^{1/2-\epsilon}-1)(1-\kappa_f)}$, for any $\epsilon > 0$.*

The simplest alternative approximation to $f$ one might conceive is the modular function $\hat{f}^m(X) \triangleq \sum_{j\in X} f(j)$ which can easily be computed by querying the $n$ values $f(j)$.

**Lemma 3.1.** *Given a monotone submodular function $f$, it holds that[2]*

$$f(X) \leq \hat{f}^m(X) = \sum_{j\in X} f(j) \leq \frac{|X|}{1+(|X|-1)(1-\kappa_f(X))}f(X) \tag{6}$$

The form of Lemma 3.1 is slightly different from Corollary 3.2. However, there is a straightforward correspondence: given $\hat{f}$ such that $\hat{f}(X) \leq f(X) \leq \alpha'(n)\hat{f}(X)$, by defining $\hat{f}'(X) = \alpha'(n)\hat{f}(X)$, we get that $f(X) \leq \hat{f}'(X) \leq \alpha'(n)f(X)$. Lemma 3.1 for the modular approximation is complementary to Corollary 3.2: First, the modular approximation is better whenever $|X| \leq \sqrt{n}$. Second, the bound in Lemma 3.1 depends on the curvature $\kappa_f(X)$ with respect to the set $X$, which is stronger than $\kappa_f$. Third, $\hat{f}^m$ is extremely simple to compute. For sets of larger cardinality, however, the ellipsoidal approximation of Corollary 3.2 provides a better approximation, in fact, the best possible one (Theorem 3.3). In a similar manner, Lemma 3.1 is tight for any modular approximation to a submodular function:

**Lemma 3.2.** *For any $\kappa > 0$, there exists a monotone submodular function $f$ with curvature $\kappa$ such that no modular upper bound on $f$ can approximate $f(X)$ to a factor better than $\frac{|X|}{1+(|X|-1)(1-\kappa_f)}$.*

The improved curvature dependent bounds immediately imply better bounds for the class of concave over modular functions used in [22, 17, 11].

**Corollary 3.4.** *Given weight vectors $w_1, \cdots, w_k \geq 0$, and a submodular function $f(X) = \sum_{i=1}^{k} \lambda_i [w_i(X)]^a, \lambda_i \geq 0$, for $a \in (0,1)$, it holds that $f(X) \leq \sum_{j \in X} f(j) \leq |X|^{1-a} f(X)$*

In particular, when $a = 1/2$, the modular upper bound approximates the sum of square-root over modular functions by a factor of $\sqrt{|X|}$.

# 4 Learning Submodular functions

We next address the problem of learning submodular functions in a PMAC setting [2]. The PMAC (Probably Mostly Approximately Correct) framework is an extension of the PAC framework [28] to allow multiplicative errors in the function values from a fixed but unknown distribution $\mathcal{D}$ over $2^V$. We are given training samples $\{(X_i, f(X_i))\}_{i=1}^{m}$ drawn i.i.d. from $\mathcal{D}$. The algorithm may take time polynomial in $n$, $1/\epsilon$, $1/\delta$ to compute a (polynomially-representable) function $\hat{f}$ that is a good approximation to $f$ with respect to $\mathcal{D}$. Formally, $\hat{f}$ must satisfy that

$$\Pr_{X_1, X_2, \cdots, X_m \sim \mathcal{D}} \left[ \Pr_{X \in \mathcal{D}}[\hat{f}(X) \leq f(X) \leq \alpha(n)\hat{f}(X)] \geq 1 - \epsilon \right] \geq 1 - \delta \qquad (7)$$

for some approximation factor $\alpha(n)$. Balcan and Harvey [2] propose an algorithm that PMAC-learns any monotone, nonnegative submodular function within a factor $\alpha(n) = \sqrt{n+1}$ by reducing the problem to that of learning a binary classifier. If we assume that we have an upper bound on the curvature $\kappa_f$, or that we can estimate it [3], and have access to the value of the singletons $f(j), j \in V$, then we can obtain better learning results with non-maximal curvature:

**Lemma 4.1.** *Let $f$ be a monotone submodular function for which we know an upper bound on its curvature and the singleton weights $f(j)$ for all $j \in V$. For every $\epsilon, \delta > 0$ there is an algorithm that uses a polynomial number of training examples, runs in time polynomial in $(n, 1/\epsilon, 1/\delta)$ and PMAC-learns $f$ within a factor of $\frac{\sqrt{n+1}}{1+(\sqrt{n+1}-1)(1-\kappa_f)}$. If $\mathcal{D}$ is a product distribution, then there exists an algorithm that PMAC-learns $f$ within a factor of $O(\frac{\log \frac{1}{\epsilon}}{1+(\log \frac{1}{\epsilon}-1)(1-\kappa_f)})$.*

The algorithm of Lemma 4.1 uses the reduction of Balcan and Harvey [2] to learn the $\kappa_f$-curve-normalized version $f^{\kappa}$ of $f$. From the learned function $\hat{f}^{\kappa}(X)$, we construct the final estimate $\hat{f}(X) \triangleq \kappa_f \hat{f}^{\kappa}(X) + (1 - \kappa_f) \sum_{j \in X} f(j)$. Theorem 3.1 implies Lemma 4.1 for this $\hat{f}(X)$. Moreover, no polynomial-time algorithm can be guaranteed to PMAC-learn $f$ within a factor of $\frac{n^{1/3-\epsilon'}}{1+(n^{1/3-\epsilon'}-1)(1-\kappa_f)}$, for any $\epsilon' > 0$ [12]. We end this section by showing how we can learn with a construction analogous to that in Lemma 3.1.

**Lemma 4.2.** *If $f$ is a monotone submodular function with known curvature (or a known upper bound) $\hat{\kappa_f}(X), \forall X \subseteq V$, then for every $\epsilon, \delta > 0$ there is an algorithm that uses a polynomial number of training examples, runs in time polynomial in $(n, 1/\epsilon, 1/\delta)$ and PMAC learns $f(X)$ within a factor of $1 + \frac{|X|}{1+(|X|-1)(1-\hat{\kappa}_f(X))}$.*

Compare this result to Lemma 4.1. Lemma 4.2 leads to better bounds for small sets, whereas Lemma 4.1 provides a better general bound. Moreover, in contrast to Lemma 4.1, here we only need an upper bound on the curvature and do not need to know the singleton weights $\{f(j), j \in V\}$. Note also that, while $\kappa_f$ itself is an upper bound of $\kappa_f(X)$, often one does have an upper bound on $\hat{\kappa}_f(X)$ if one knows the function class of $f$ (for example, say concave over modular). In particular, an immediate corollary is that the class of concave over modular functions $f(X) = \sum_{i=1}^k \lambda_i [w_i(X)]^a, \lambda_i \geq 0$, for $a \in (0,1)$ can be learnt within a factor of $\min\{\sqrt{n+1}, 1 + |X|^{1-a}\}$.

## 5 Constrained submodular minimization

Next, we apply our results to the minimization of submodular functions under constraints. Most algorithms for constrained minimization use one of two strategies: they apply a convex relaxation [10, 16], or they optimize a surrogate function $\hat{f}$ that should approximate $f$ well [6, 8, 16]. We follow the second strategy and propose a new, widely applicable curvature-dependent choice for surrogate functions. A suitable selection of $\hat{f}$ will ensure theoretically optimal results. Throughout this section, we refer to the optimal solution as $X^* \in \operatorname{argmin}_{X \in \mathcal{C}} f(X)$.

**Lemma 5.1.** *Given a submodular function $f$, let $\hat{f}_1$ be an approximation of $f$ such that $\hat{f}_1(X) \leq f(X) \leq \alpha(n)\hat{f}_1(X)$, for all $X \subseteq V$. Then any minimizer $\widehat{X}_1 \in \operatorname{argmin}_{X \in \mathcal{C}} \hat{f}(X)$ of $\hat{f}$ satisfies $f(\widehat{X}) \leq \alpha(n)f(X^*)$. Likewise, if an approximation of $f$ is such that $f(X) \leq \hat{f}_2(X) \leq \alpha(X)f(X)$ for a set-specific factor $\alpha(X)$, then its minimizer $\tilde{X}_2 \in \operatorname{argmin}_{X \in \mathcal{C}} \hat{f}_2(X)$ satisfies $f(\widehat{X}_2) \leq \alpha(X^*)f(X^*)$. If only $\beta$-approximations[4] are possible for minimizing $\hat{f}_1$ or $\hat{f}_2$ over $\mathcal{C}$, then the final bounds are $\beta\alpha(n)$ and $\beta\alpha(X^*)$ respectively.*

For Lemma 5.1 to be practically useful, it is essential that $\hat{f}_1$ and $\hat{f}_2$ be efficiently optimizable over $\mathcal{C}$. We discuss two general curvature-dependent approximations that work for a large class of combinatorial constraints. In particular, we use Theorem 3.1: we decompose $f$ into $f^\kappa$ and a modular part $f^m$, and then approximate $f^\kappa$ while retaining $f^m$, i.e., $\hat{f} = \hat{f}^\kappa + f^m$. The first approach uses a simple modular upper bound (MUB) and the second relies on the Ellipsoidal approximation (EA) we used in Section 3.

**MUB:** The simplest approximation to a submodular function is the modular approximation $\hat{f}^m(X) \triangleq \sum_{j \in X} f(j) \geq f(X)$. Since here, $\hat{f}^\kappa$ happens to be equivalent to $f^m$, we obtain the overall approximation $\hat{f} = \hat{f}^m$. Lemmas 5.1 and 3.1 directly imply a set-dependent approximation factor for $\hat{f}^m$:

**Corollary 5.1.** *Let $\widehat{X} \in \mathcal{C}$ be a $\beta$-approximate solution for minimizing $\sum_{j \in X} f(j)$ over $\mathcal{C}$, i.e. $\sum_{j \in \widehat{X}} f(j) \leq \beta \min_{X \in \mathcal{C}} \sum_{j \in X} f(j)$. Then*

$$f(\hat{X}) \leq \frac{\beta|X^*|}{1 + (|X^*| - 1)(1 - \kappa_f(X^*))} f(X^*). \tag{8}$$

Corollary 5.1 has also been shown in [13]. Similar to the algorithms in [13], MUB can be extended to an iterative algorithm yielding performance gains in practice. In particular, Corollary 5.1 implies improved approximation bounds for practically relevant concave over modular functions, such as those used in [17]. For instance, for $f(X) = \sum_{i=1}^k \sqrt{\sum_{j \in X} w_i(j)}$, we obtain a worst-case approximation bound of $\sqrt{|X^*|} \leq \sqrt{n}$. This is significantly better than the worst case factor of $|X^*|$ for general submodular functions.

**EA:** Instead of employing a modular upper bound, we can approximate $f^\kappa$ using the construction by Goemans et al. [8], as in Corollary 3.2. In that case, $\hat{f}(X) = \kappa_f \sqrt{w^{f^\kappa}(X)} + (1 - \kappa_f)f^m(X)$ has a special form: a weighted sum of a concave function and a modular function. Minimizing such a function over constraints $\mathcal{C}$ is harder than minimizing a merely modular function, but with the algorithm in [24] we obtain an FPTAS[5] for minimizing $\hat{f}$ over $\mathcal{C}$ whenever we can minimize a nonnegative linear function over $\mathcal{C}$.

**Corollary 5.2.** *For a submodular function with curvature $\kappa_f < 1$, algorithm EA will return a solution $\widehat{X}$ that satisfies*

$$f(\widehat{X}) \leq O\left(\frac{\sqrt{n}\log n}{(\sqrt{n}\log n - 1)(1 - \kappa_f) + 1}\right) f(X^*). \tag{9}$$

Next, we apply the results of this section to specific optimization problems, for which we show (mostly tight) curvature-dependent upper and lower bounds. We just state our main results; a more extensive discussion along with the proofs can be found in [12].

**Cardinality lower bounds (SLB).** A simple constraint is a lower bound on the cardinality of the solution, i.e., $\mathcal{C} = \{X \subseteq V : |X| \geq k\}$. Svitkina and Fleischer [27] prove that for monotone submodular functions of arbitrary curvature, it is impossible to find a polynomial-time algorithm with an approximation factor better than $\sqrt{n/\log n}$. They show an algorithm which matches this approximation factor. Corollaries 5.1 and 5.2 immediately imply *curvature-dependent* approximation bounds of $\frac{k}{1+(k-1)(1-\kappa_f)}$ and $O(\frac{\sqrt{n}\log n}{1+(\sqrt{n}\log n-1)(1-\kappa_f)})$. These bounds are improvements over the results of [27] whenever $\kappa_f < 1$. Here, MUB is preferable to EA whenever $k$ is small. Moreover, the bound of EA is tight up to poly-log factors, in that no polynomial time algorithm can achieve a general approximation factor better than $\frac{n^{1/2-\epsilon}}{1+(n^{1/2-\epsilon}-1)(1-\kappa_f)}$ for any $\epsilon > 0$.

In the following problems, our ground set $V$ consists of the set of edges in a graph $\mathcal{G} = (\mathcal{V}, \mathcal{E})$ with two distinct nodes $s, t \in V$ and $n = |\mathcal{V}|$, $m = |\mathcal{E}|$. The submodular function is $f : 2^{\mathcal{E}} \to \mathbb{R}$.

**Shortest submodular s-t path (SSP).** Here, we aim to find an s-t path $X$ of minimum (submodular) length $f(X)$. Goel et al. [6] show a $O(n^{2/3})$-approximation with matching curvature-independent lower bound $\Omega(n^{2/3})$. By Corollary 5.1, the curvature-dependent worst-case bound for MUB is $\frac{n}{1+(n-1)(1-\kappa_f)}$ since any minimal s-t path has at most $n$ edges. Similarly, the factor for EA is $O(\frac{\sqrt{m}\log m}{1+(\sqrt{m}\log m-1)(1-\kappa_f)})$. The bound of EA will be tighter for sparse graphs while MUB provides better results for dense ones. Our curvature-dependent lower bound for SSP is $\frac{n^{2/3-\epsilon}}{1+(n^{2/3-\epsilon}-1)(1-\kappa_f)}$, for any $\epsilon > 0$, which reduces to the result in [6] for $\kappa_f = 1$.

**Minimum submodular s-t cut (SSC):** This problem, also known as the cooperative cut problem [16, 17], asks to minimize a monotone submodular function $f$ such that the solution $X \subseteq \mathcal{E}$ is a set of edges whose removal disconnects $s$ from $t$ in $\mathcal{G}$. Using curvature refines the We can also show a lower bound of [16] to $\frac{n^{1/2-\epsilon}}{1+(n^{1/2-\epsilon}-1)(1-\kappa_f)}$, for any $\epsilon > 0$. Corollary 5.1 implies an approximation factor of $O(\frac{\sqrt{m}\log m}{(\sqrt{m}\log m-1)(1-\kappa_f)+1})$ for EA and a factor of $\frac{m}{1+(m-1)(1-\kappa_f)}$ for MUB, where $m = |\mathcal{E}|$ is the number of edges in the graph. Hence the factor for EA is tight for sparse graphs. Specifically for cut problems, there is yet another useful surrogate function that is exact on local neighborhoods. Jegelka and Bilmes [16] demonstrate how this approximation may be optimized via a generalized maximum flow algorithm that maximizes a *polymatroidal network flow* [20]. This algorithm still applies to the combination $\hat{f} = \kappa_f \hat{f}^\kappa + (1 - \kappa_f)f^m$, where we only approximate $f^\kappa$. We refer to this approximation as Polymatroidal Network Approximation (PNA).

**Corollary 5.3.** *Algorithm PNA achieves a worst-case approximation factor of $\frac{n}{2+(n-2)(1-\kappa_f)}$ for the cooperative cut problem.*

For dense graphs, this factor is theoretically tighter than that of the EA approximation.

**Minimum submodular spanning tree (SST).** Here, $\mathcal{C}$ is the family of all spanning trees in a given graph $\mathcal{G}$. Such constraints occur for example in power assignment problems [30]. Goel et al. [6] show a curvature-independent optimal approximation factor of $O(n)$ for this problem. Corollary 5.1 refines this bound to $\frac{n}{1+(n-1)(1-\kappa_f)}$ when using MUB; Corollary 5.2 implies a slightly worse bound for EA. We also show that the bound of MUB is tight: no polynomial-time algorithm can guarantee a factor better than $\frac{n^{1-\epsilon}}{1+(n^{1-\epsilon}-1)(1-\kappa_f)+\delta\kappa_f}$, for any $\epsilon, \delta > 0$.

**Minimum submodular perfect matching (SPM):** Here, we aim to find a perfect matching in a graph that minimizes a monotone submodular function. Corollary 5.1 implies that an MUB approximation will achieve an approximation factor of at most $\frac{n}{2+(n-2)(1-\kappa_f)}$. Similar to the spanning tree case, the bound of MUB is also tight [12].

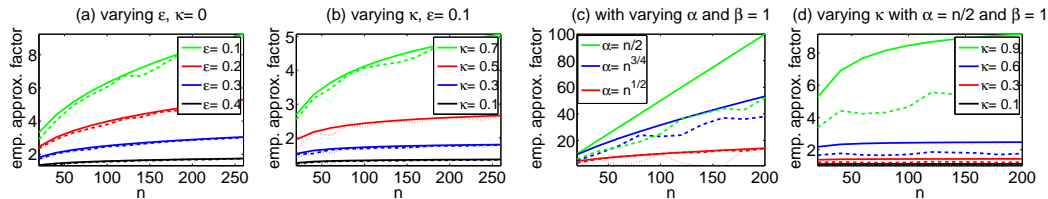

Figure 1: Minimization of $g_\kappa$ for cardinality lower bound constraints. (a) fixed $\kappa = 0$, $\alpha = n^{1/2+\epsilon}$, $\beta = n^{2\epsilon}$ for varying $\epsilon$; (b) fixed $\epsilon = 0.1$, but varying $\kappa$; (c) different choices of $\alpha$ for $\beta = 1$; (d) varying $\kappa$ with $\alpha = n/2$, $\beta = 1$. Dashed lines: MUB, dotted lines: EA, solid lines: theoretical bound. The results of EA are not visible in some instances since it obtains a factor of 1.

## 5.1 Experiments

We end this section by empirically demonstrating the performance of MUB and EA and their precise dependence on curvature. We focus on cardinality lower bound constraints, $\mathcal{C} = \{X \subseteq V : |X| \geq \alpha\}$ and the "worst-case" class of functions that has been used throughout this paper to prove lower bounds, $f^R(X) = \min\{|X \cap \bar{R}| + \beta, |X|, \alpha\}$ where $\bar{R} = V \setminus R$ and $R \subseteq V$ is random set such that $|R| = \alpha$. We adjust $\alpha = n^{1/2+\epsilon}$ and $\beta = n^{2\epsilon}$ by a parameter $\epsilon$. The smaller $\epsilon$ is, the harder the problem. This function has curvature $\kappa_f = 1$. To obtain a function with specific curvature $\kappa$, we define $f^R_\kappa(X) = \kappa f(X) + (1 - \kappa)|X|$ as in Equation (3).

In all our experiments, we take the average over 20 random draws of $R$. We first set $\kappa = 1$ and vary $\epsilon$. Figure 1(a) shows the empirical approximation factors obtained using EA and MUB, and the theoretical bound. The empirical factors follow the theoretical results very closely. Empirically, we also see that the problem becomes harder as $\epsilon$ decreases. Next we fix $\epsilon = 0.1$ and vary the curvature $\kappa$ in $f^R_\kappa$. Figure 1(b) illustrates that the theoretical and empirical approximation factors improve significantly as $\kappa$ decreases. Hence, much better approximations than the previous theoretical lower bounds are possible if $\kappa$ is not too large. This observation can be very important in practice. Here, too, the empirical upper bounds follow the theoretical bounds very closely.

Figures 1(c) and (d) show results for larger $\alpha$ and $\beta = 1$. In Figure 1(c), as $\alpha$ increases, the empirical factors improve. In particular, as predicted by the theoretical bounds, EA outperforms MUB for large $\alpha$ and, for $\alpha \geq n^{2/3}$, EA finds the optimal solution. In addition, Figures 1(b) and (d) illustrate the theoretical and empirical effect of curvature: as $n$ grows, the bounds saturate and approximate a constant $1/(1 - \kappa)$ – they do not grow polynomially in $n$. Overall, we see that the empirical results quite closely follow our theoretical results, and that, as the theory suggests, curvature significantly affects the approximation factors.

## 6 Conclusion and Discussion

In this paper, we study the effect of curvature on the problems of approximating, learning and minimizing submodular functions under constraints. We prove tightened, curvature-dependent upper bounds with almost matching lower bounds. These results complement known results for submodular maximization [3, 29]. Given that the functional form and effect of the submodularity ratio proposed in [5] is similar to that of curvature, an interesting extension is the question of whether there is a single unifying quantity for both of these terms. Another open question is whether a quantity similar to curvature can be defined for subadditive functions, thus refining the results in [1] for learning subadditive functions. Finally it also seems that the techniques in this paper could be used to provide improved curvature-dependent regret bounds for constrained online submodular minimization [15].

**Acknowledgments:** Special thanks to Kai Wei for pointing out that Corollary 3.4 holds and for other discussions, to Bethany Herwaldt for reviewing an early draft of this manuscript, and to the anonymous reviewers. This material is based upon work supported by the National Science Foundation under Grant No. (IIS-1162606), a Google and a Microsoft award, and by the Intel Science and Technology Center for Pervasive Computing. Stefanie Jegelka's work is supported by the Office of Naval Research under contract/grant number N00014-11-1-0688, and gifts from Amazon Web Services, Google, SAP, Blue Goji, Cisco, Clearstory Data, Cloudera, Ericsson, Facebook, General Electric, Hortonworks, Intel, Microsoft, NetApp, Oracle, Samsung, Splunk, VMware and Yahoo!.

## Footnotes

[1] A polymatroid function is a monotone increasing, nonnegative, submodular function satisfying $f(\emptyset) = 0$.

[2]In [12], we show this result with a stronger notion of curvature: $\hat{\kappa}_f(X) = 1 - \frac{\sum_{j\in X} f(j|X\backslash j)}{\sum_{j\in X} f(j)}$.

[3]note that $\kappa_f$ can be estimated from a set of $2n + 1$ samples $\{(j, f(j))\}_{j \in V}$, $\{(V, f(V))\}$, and $\{(V \backslash j, f(V \backslash j))\}_{j \in V}$ included in the training samples

[4]A $\beta$-approximation algorithm for minimizing a function $g$ finds set $X : g(X) \leq \beta \min_{X \in \mathcal{C}} g(X)$

[5]The FPTAS will yield a $\beta = (1 + \epsilon)$-approximation through an algorithm polynomial in $\frac{1}{\epsilon}$.

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
