[Reviews · NeurIPS 2013]

Submitted by Assigned_Reviewer_5

Curvature and optimal algorithms for learning and minimizing submodular functions

This paper explores the impact of a notion of function curvature on the approximability of problems involving submodular functions. While it's not the first work to use this notion of curvature to improve bounds for submodular functions (for submodular maximization), this paper covers more settings. In particular, the problems covered are:
- efficiently finding (and representing) a function f that approximates the submodular function
- PAC-style learning
- constrained function minimization

The paper makes the case that many practically interesting functions have small curvature and thus admit better approximations that previous bounds would indicate.

The main approach to solving the approximation problem is to separately approximate the cured part of the underlying function. The approach to the second problem is to use the Balcan-Harvey PAC-style learning algorithm and apply the approximation result. The approach to the third problem is less black-box than the previous two, but gives a means of choosing an approximation of the original function to analyze; then they can apply the previously mentioned results to understand how good an approximation this gives.

Finally, there is a brief experimental section to demonstrate that for functions with small curvature (for the "worst case" given that curvature), the empirical constrained minimization closely tracks the approximation predicted.

These are important problems, with lots of different applications, and it's nice to see some general tools for understanding what affects (in)approximability.

It would have been helpful to have more discussion of the distinction between the notion of curvature used here and the submodularity ratio (is there a way to compare their usefulness in improving approximation ratios?)

It is a bit difficult to digest the new results, since the paper doesn't provide a succinct comparison with existing worst-case bounds (for functions with maximum curvature). I realize of course that there are a lot of other comparisons with previous work one could make (for other specific classes of functions), but this particular comparison seems that it would help the reader understand the impact of taking curvature into account.
Summary: These are important problems, with lots of different applications, and it's nice to see some general tools for understanding what affects (in)approximability.

Submitted by Assigned_Reviewer_8

The authors use a property of monotone sub-modular functions, the curvature, that ranges from 0 for modular function to 1 for functions such as f(S) = min(|S|, k). They show that any sub-modular function can be presented as a sum of a modular function and a 1-curved sub-modular function. This observation allow the authors to show that many results known about sub-modular functions can be improved by taking into account the curvature parameter.
Sub-modular functions are very “popular” in machine learning study in recent years, hence contribution in this field is of importance. The observation made by the authors in very nice and allows some to apply it to a large list of results. The article left me wishing the author could have gone in greater depth to discuss ways to compute/approximate the curvature. Since the authors focused on optimization problems where the function is provided using an oracle, how can one estimate the curvature to make use of the improved properties the authors reported.
The presentation could use some work in making the statements in this paper more accurate:
1. in (1): how does one define k_f if there exists i such that f(i) = 0?
2. For example, after (2) the authors claim that $ k f^k = 1$ but this does not hold when k = 0.
3. Theorem 3.1 should state the assumption that k_f < 1 instead of only presenting it in the text before the statement of the theorem
4. Corollary 3.2 uses the term “ellipsoidal approximation” which is not defined
5. In line 291 the authors use the term “$\beta$ approximation” which is not defined

Summary: The authors present a way to break monotone sub-modular function into a sum of modular function and a 1-curved monotone sub-modular function where the coefficients depend on the curvature. This allows them to achieve better bounds on many optimization problems if the curvature is known. However, they do not show way to compute or approximate the curvature for functions that are provided by an oracle.


Submitted by Assigned_Reviewer_9

This paper gives new bounds and algorithms for problems involving submodular functions including approximation, PMAC learning and optimization. The bounds improve on current bounds and are often matched by similar lower bounds. The key additional in feature is the curvature of a submodular function. Current upper bounds usually involved worst case curvature but it apears that a lot can be gained when curvature is small - the new bounds improve significantly in this case.

The paper presents many results and the problems are well motivated.
Summary: new improved bounds for several well motivated problems involving submodular functions

Submitted by Assigned_Reviewer_11

I refer to the supplementary paper (which is the superset of the main paper).

The paper refines existing bounds on approximability of several optimization problems involving submodular set functions. This is done by incorporating
the curvature of the submodular function into the bounds. Since existing bounds were derived for the worst case (curvature=1), this makes the bounds more
optimistic whenever the function has curvature less than 1.

In particular, the bounds are refined for three problems:
- approximating a submodular function,
- learning a submodular function from samples
- minimizing a submod function subject to several types of constraints (e.g., the feasible set is the set of all cuts in a graph, or the set of all sets
with a bounded cardinality).

The paper is very cleanly written, given the complexity of the topic, many works are cited. It is apparent that the authors are good mathematicians,
experts in approximability.

I did not find any major technical problems. However, let me note that my knowledge of the topic may not be deep enough to assess this. It would be
difficult for me to verify all the proofs in detail.
For the same reason, I cannot completely guarantee than some the presented results have not been published somewhere before.

The paper has little enough overlap with paper 1147 to justify simultaneous publications.

However, it seems to me that the text would be better suited for a mathematical journal than NIPS. First, there is not enough pages at NIPS. The problems
considered are of enough general interest independent on machine learning, to be suited for a mathematical journal. Second, the NIPS audience are not
mathematicians but more often engineers, thus they may not esily understand the proofs.

Minor comments (refer to the supplement):

- Proof of Lemma 2.1: Why is function f monotonic? This does not follow
from the assumption of the lemma.

- Notation in eq. (5) clashes with that in eq. (3)

- Line 199: alpha_1(n)=O(sqrt{n}log{n})

- Lemma 5.1: X^* has not been defined (though obvious)

- Eq. (28): overline{R} has not been defined
Summary: Clear contribution, very cleanly written. Good paper, I recommend acceptation.
However, the text may be better suited for a mathematical journal than NIPS.
Author Feedback

Author rebuttal: We would like to thank all the reviewers for their time and reviews. We address the issues pointed out by each reviewer separately below.

Assigned_Reviewer_5:
The curvature and submodularity ratio capture different aspects of set functions -- the first is a distance of a monotone submodular function to modularity, while the second is a notion of 'distance' to submodularity. We discuss this to some extent in the extended version (lines 137 - 145). It is an interesting question, however, if there is a unifying quantity that captures both notions (we point this out in
the discussion in the paper, lines 429-430). We believe both curvature and the submodularity ratio are useful, but curvature is easy to compute and (as shown in this paper) is very widely applicable. If accepted, we'll offer a bit more discussion on the relative utility of each measure.


We compare our new theoretical results with the existing worst case ones in different parts of the paper. For the problems of approximating and learning submodular functions, we discuss this in lines 156 - 176. For constrained minimization, we discuss these improvements, in many cases, in the specific section that deals with these constraints. We will, however, make sure to add this discussion in every section (and also possibly in Table 1), if the paper gets
accepted.

Assigned Reviewer_8:
Computation of the curvature: As evident from eqn. 1, the curvature can indeed be computed very easily for any submodular function (in just O(n) oracle calls). The same holds for computing the curvature with respect to a given set. Hence these bounds can directly be obtained given oracle access to a submodular function. Note that this is distinct from the submodularity ratio, which is hard to compute.

Minor issues: (We denote \kappa by k, for simplicity)
1) It is true that k_f will not be defined if f(i) = 0, because this also implies that f(i | V / i) = 0 (because of submodularity) and we obtain 0/0. In such a case however, we can safely remove element i from the ground set, since for every set X, f(i | X) = 0, and including or excluding i does not make any difference to the cost function.
Hence we can assume w.l.o.g that f(i) > 0 for all i in V. We will add this to the paper, if accepted.

2) If the curvature k_f = 0, we assume the curve-normalized part f^{\kappa} = 0 since the numerator in (2) is zero, and indeed in this case, it doesn't make sense to define k_{f^k} = 1. However, in all other cases (i.e when f^k \neq 0), it holds that k_{f^k} = 1 (Note the that the quantity in lines 128-129 of the main paper is k_{f^k} and not k f^k as pointed out in the second point by Assigned Reviewer_8). We will clarify this in the paper, if accepted.

3 - 5) We will add all these suggestions into the main paper, if it is accepted.

Assigned_reviewer_9:
We thank you for your encouraging review.